# Development and validation of a dual language needs assessment tool for people living with colorectal cancer (NeAT-CC)

Nur Nadiatul Asyikin Bujang[1,2], Yek Ching Kong[1], Muthukkumaran Thiagarajan[3], April Camilla Roslani[4], Muhammad Radzi Abu Hassan[2,5,6], Matin Mellor Abdullah[7], Mehesinder Singh[8], Wan Zamaniah Wan Ishak[9], Awang Bulgiba[1], Mahmoud Danaee[1], Nirmala Bhoo-Pathy[1]*

1 Centre for Epidemiology and Evidence-Based Practice, Department of Social and Preventive Medicine, Faculty of Medicine, Universiti Malaya, Kuala Lumpur, Malaysia, 2 Ministry of Health, Putrajaya, Wilayah Persekutuan Putrajaya, Malaysia, 3 Department of Radiotherapy and Oncology, Hospital Kuala Lumpur, Kuala Lumpur, Malaysia, 4 Department of Surgery, Faculty of Medicine, Universiti Malaya, Kuala Lumpur, Malaysia, 5 Department of Internal Medicine, Hospital Sultanah Bahiyah, Alor Setar, Kedah, Malaysia, 6 Clinical Research Centre, Hospital Sultanah Bahiyah, Alor Setar, Kedah, Malaysia, 7 Subang Jaya Medical Centre, Subang Jaya, Malaysia, 8 Colorectal Cancer Survivorship Society Malaysia (CORUM), Pantai Hospital Kuala Lumpur, Kuala Lumpur, Malaysia, 9 Clinical Oncology Unit, Faculty of Medicine, Universiti Malaya, Kuala Lumpur, Malaysia

* ovenjjay@gmail.com, nirmala.bhoopathy@ummc.edu.my

## Abstract

### Background

Needs assessment tools may guide optimization of clinical services to be more patient-centered. As needs of patients living with and beyond colorectal cancer (CRC) may also be influenced by socio-cultural backgrounds and healthcare ecosystems, we developed and validated a needs assessment questionnaire for CRC in a multi-ethnic, low-and middle-income setting.

### Methods

The study methodology was guided by the COSMIN checklist. Items generation was based on findings from independent qualitative inquiries with patients, input from cancer stakeholders, and literature review. Following translation into Malay language, content and face validation were undertaken. The tool was administered to 300 individuals living with and beyond CRC. Exploratory factor analysis (EFA) and confirmatory factor analysis (CFA) were performed. Criterion validity was assessed using EORTC QLQ-C30 and QLQ-CR29 questionnaires.

### Results

The 48-item bilingual needs assessment tool for colorectal cancer (NeAT-CC) encompassed six domains of needs, namely (i) diagnosis, (ii) psychosocial and information,

**Data availability statement:** The data that support the findings of this study are not publicly available due to confidentiality and data protection requirements imposed by the National Medical Research Register (NMRR) and the Ministry of Health Malaysia. As stipulated in the study approval, all data must be kept strictly confidential and used solely for the purposes of this study. Requests for data access may be made to the Medical Research and Ethics Committee (MREC), Ministry of Health Malaysia, subject to appropriate approvals. The NMRR can be contacted at Phone: +603-3362 8205/8079/8898 or Email: nmrr@moh.gov.my. Alternatively, requests may be directed to the corresponding author at ovenjjay@gmail.com or the first author at nadiatulasyikin@moh.gov.my.

**Funding:** This study was supported by the Long-Term Research Grant Scheme (LR001C-2019) from the Ministry of Higher Education, Malaysia. The funders had no role in the study design, data collection, data analysis, or manuscript preparation and publication.

**Competing interests:** The authors have no financial and non-financial conflict of interest to declare.

(iii) healthcare, (iv) practical and living with cancer, (v) financial and (vi) employment. Cronbach's alpha was above 0.70 for all domains, indicating good internal consistency. CFA also demonstrated acceptable convergent and divergent validity with composite reliability >0.70 and Heterotrait–Monotrait index <0.90 for all constructs. Criterion validity was established given the significant correlation with quality of life. The NeAT-CC was easily understandable, took 15–20 minutes for completion and may be self-administered.

## Conclusions

Utilization of NeAT-CC may facilitate optimization of supportive and survivorship care services following CRC in local settings. The tool has wider potential for adaptation in other multi-ethnic and/or low and middle-income settings.

---

## 1. Introduction

Recent global estimates suggest that survival following colorectal cancer (CRC) has been increasing in many parts of the world, including in some low and middle-income countries such as Malaysia [1]. The rise in number of patients who are surviving colorectal cancer (CRC) for extended periods of time has led to increased recognition of the importance of understanding and assessing their specific care needs [2]. A driving impetus here is the growing global movement for delivery of patient-centered care, which emphasizes that the needs, values, and preferences of patients are consistently taken into the planning of care delivery across the cancer continuum [3].

Notably, prior evidence suggests that patients whose care needs are met tend to exhibit higher self-efficacy and self-management, greater adherence to health promotion approaches and experience less suffering and lower symptom burden, leading to improved overall wellbeing [4,5]. From the health system's perspective, detailed knowledge about the issues and concerns of people living with and beyond CRC, their care needs and the extent to which these are met by current services are critical in guiding the allocation of scarce healthcare resources, and planning the delivery of high-quality supportive care and survivorship care.

Commonly used tools such as the Supportive Care Need Survey (SCNS-SF) [6], Cancer Survivors' Unmet Needs Measure (CaSUN) [7], Survivors' Unmet Needs Survey (SUNS) [8], and the Comprehensive Needs Assessment Tool in Cancer (CNAT) developed in Korea [9] were reviewed. However, most of the currently available needs assessment tools for cancer were developed in affluent settings [9–11] with more advanced healthcare systems that may not necessarily be applicable to resource-limited settings [9]. Apart from clinical factors and healthcare ecosystems, care needs may also be influenced by the patient's sociocultural background, particularly in multi-ethnic settings [9,12,13]. Prior reviews also suggest that many of the existing tools do not comprehensively cover all domains of needs following cancer such as social needs, or do not demonstrate adequate psychometric properties [14]. While certain needs following CRC have been reported to be similar with other

cancers, there are some unique challenges, such as dealing with stoma that necessitates development of a disease-specific tool [14,15]. Needs assessment tools tailored to these unique challenges are essential to ensure that survivors' concerns are effectively identified and addressed [14].

In sum, there are compelling justifications for development of a more comprehensive tool to measure needs following CRC in diverse populations. We sought to develop a dual language (English and Malay) needs assessment questionnaire for people living with and beyond CRC in a multi-ethnic, low and middle-income setting.

## 2. Methods

In the current study, need was defined as the level of service or support that an individual perceived as necessary to achieve optimal well-being following CRC [16]. People living with and beyond CRC in the context of present study included individuals diagnosed at least one-month prior to study recruitment who were either waiting for cancer treatment or under active treatment, or individuals who had completed treatment (irrespective of time since diagnosis) [17]. Recruitment was carried out from 12 April 2019 to 28 November 2020.

Study methodology was guided by the COnsensus-based Standards for the selection of health status Measurement INstruments (COSMIN) checklist [18], comprising three major phases; a) item development, b) scale development and c) scale evaluation [19]. Ethical approvals were obtained from the Medical Research and Ethics Committee [NMRR-18-3687-44522], University Malaya Medical Centre Medical Ethics Committee [20181016-6757] and Subang Jaya Medical Centre [201907.1]. Written and informed consent was obtained from all the study participants.

### 2.1. Phase I: Item development

A co-design approach was adopted, where a wide range of cancer stakeholders were actively involved in developing the items for the needs assessment tool.

**2.1.1. Item generation and domain identification.** Generation of items in the initial draft of the questionnaire was guided by findings from independent qualitative inquiries that were previously undertaken by the study team in the local setting among people living with and beyond cancer. Details can be found elsewhere [20–25]. Concurrently, a literature review of the existing needs assessment tools for cancer was also conducted to screen for items [6,7,9–11,26]. Given that prior qualitative inquiries and needs assessment tools were from the pre COVID-19 era, all the items were reviewed by a diverse group of cancer stakeholders including individuals living with and beyond CRC, caregivers, cancer advocates and healthcare professionals (oncologists, surgeons, nurses, psychologists, counsellors, public health physicians) to ensure that contemporary issues were not missed. Shortlisted items were categorized according to subconstructs of needs (domains) based on consensus from the stakeholders, and in accordance with the supportive care framework proposed by Fitch 2008 [27].

**2.1.2. Content validation.** Expert validation and face validation were conducted to establish that the generated items were relevant to the needs of people living with and beyond CRC and that key items have not been missed. A panel of seven content experts comprising healthcare professionals, a cancer researcher, a patient advocate and a patient living with CRC were convened to provide feedback on the generated items. The panel members were asked to rate the items using a four-point scale (1: not relevant, 2: not important, 3: relevant and 4: important), where a rating of three or four indicated agreement. Two rounds of content validation were conducted. In the first round, the acceptable cut-off for the item content validity index (I-CVI) was >= 0.83, while in the second round, the cut-off was >= 0.86 [28]. Kappa was also computed for individual items where a value above 0.74 was considered excellent [29]. Scale content validity index (S-CVI) was computed by calculating the average I-CVI of each domain, in which a cutoff value ≥ 0.90 was deemed acceptable [29].

**2.1.3. Readability and translation.** The Flesch–Kincaid reading grade (FK), which was calculated using an online tool was used to determine the readability of the (English) items in the questionnaire [30]. All items were forward translated to the Malay

language by two bilingual members of the study team. The dual language items were then reviewed by patients and cancer advocates who were proficient in both languages. Their feedback was used to revise the Malay items to improve clarity and comprehensibility. In the next step, the Malay version was reverse translated into English by two other independent translators who were also proficient in both languages. The back-translated questionnaire in English was compared with the original questionnaire. After item harmonisation and consensus were achieved, the final version of the questionnaire covered dual languages (Malay and English). Grammar and spelling were checked prior to cognitive debriefing.

## 2.2. Phase II and III (Psychometric evaluation): Scale development and evaluation

### 2.2.1. Study population, data collection and study instruments.
Malaysians above the age of 18 years who were living with and beyond CRC, diagnosed with any stage of CRC at least one month prior to recruitment, with or without stoma, were recruited. Individuals who were clinically unfit, as well as those who were unable to converse in English or Malay were excluded. Malay is the national language and is widely spoken across all major ethnic groups in Malaysia, while English serves as the second language and is commonly used in healthcare and research settings.

Respondents were recruited from multiple study sites serving ethnically diverse catchment populations across Klang Valley, the largest urban agglomeration in Malaysia. These included a public university hospital, several Ministry of Health (MOH)-owned hospitals, a private hospital and a local non-governmental organization (NGO) for CRC. Our sampling strategy was expected to enable adequate representation of patients from diverse ethnic and socioeconomic backgrounds in Malaysia. Notably, Malaysia has a pluralistic health system, where healthcare is funded by various means in the public and private hospitals, with differences in the fee structures and package of subsidized care even between the public system leading to differences in the socioeconomic backgrounds of patients [31].

Eligible respondents were identified during their routine outpatient visits or day care attendance in the hospital settings, whereas in the NGO setting, they were identified by a liaison officer from within the organization. After identification, they were then randomly selected using an online random number generator. Several recruitment methods were employed, including face-to-face encounters, online recruitment by enumerators, and through an NGO representative, to ensure continued data collection during the COVID-19 pandemic. Face-to-face recruitment occurred when restrictions were lifted, while online methods were used during lockdowns to minimize contact with high-risk cancer survivors. The recruitment method was based on the patient's location and preference, with informed consent obtained after participants were briefed and provided with a participant information sheet.

Given that for exploratory factor analysis (EFA), a minimum sample size of 100 [32], or three participants per item were recommended [33], 155 participants were deemed adequate. For confirmatory factor analysis (CFA), a sample size of 145 was considered appropriate based on sample size estimation using G-power software [34].

The NeAT-CC required study respondents to rate the magnitude of their needs based on a six-point Likert scale (ranging from zero, indicating no need, up to five, which denoted very high needs). The European Organisation for Research and Treatment of Cancer Quality of Life Core 30 (EORTC QLQ-C30) [35,36], as well as the colorectal cancer supplement (EORTC QLQ-C29) [37,38] quality of life questionnaires was also administered. Data on sociodemographic factors were collected using a brief data collection form. Clinical details were retrieved from the hospital, and/or patient-held records.

### 2.2.2. Cognitive pretest and test-retest.
Cognitive pre-testing was conducted with 20 people living with and beyond CRC. Purposive sampling was undertaken to recruit patients at various clinical stages of CRC (early, advanced), treatment status (active treatment, not on active treatment), ethnicity (Malay, Chinese, Indians), and type of hospital (public, private). Respondents were asked to complete the questionnaire, and their feedback was used to further revise the questionnaire to improve feasibility, acceptability, and item relevance.

Test-retest reliability was assessed among 30 participants who were conveniently sampled to complete the questionnaire at two time points (two weeks apart). Intra-rater reliability for each item was determined using weighted kappa with linear weightage via vassarstat.net online software. A cut-off value of>=0.4 was used [39].

**2.2.3. Exploratory Factor Analysis (EFA).** Exploratory factor analysis was conducted via the Factor program, using Unweighted Least Squares (ULS) [40] and the oblique Promin rotation method. The Kaiser-Meyer-Olkin test (KMO; cutoff ≥ 0.60) and Bartlett's test of sphericity (p value <0.05) [41] were conducted to determine sampling adequacy. Parallel analysis was performed to determine the appropriate number of domains to extract, and the pattern matrix was examined to assess the factor to which the items loaded. Items with a factor loading of < 0.32, as well as those with cross-loadings that did not differ by at least 0.15, were removed from the model [42,43].

**2.2.4. Reliability.** The reliability of the questionnaire was determined via examination of Cronbach's alpha, inter-item correlation, and corrected item-total correlation (CITC) using IBM Statistical Package for Social Science (SPSS). Values of at least 0.30 for the CITC and 0.70 for Cronbach's alpha were deemed acceptable [19].

**2.2.5. Confirmatory factor analysis (CFA).** Confirmatory factor analysis was conducted via the SmartPLS software 3.2.9 using the partial least squares structural equation modelling (PLS-SEM) method. This entailed measurement model assessment for a reflective construct, which included assessment of factor loadings (cut-off of ≥ 0.50) [42], Cronbach's alpha and composite reliability (CR) (>0.70) [42,44]. To assess convergent validity, the average variance extracted (AVE) was examined. An AVE of above 0.4 was considered acceptable if the CR of the domain was above 0.60 [45]. Discriminant validity was determined by assessing the Heterotrait-Monotrait ratio (HTMT, < 0.90 [46], and cross-loading of indicators (< 0.10) [47]. During the CFA phase, we evaluated the factor structure established through EFA without contrasting alternative models. This decision was predicated on the robust theoretical framework and empirical precision of the EFA solution. The EFA demonstrated a clear and interpretable structure characterized by high item loadings and minimal cross-loadings, which informed the development of the CFA model.

**2.2.6. Criterion validity.** Given the lack of a gold standard questionnaire for needs assessment in people living with and beyond cancer, predictive criterion validity was assessed by using health-related quality of life [14,48,49]. The English and Malay versions of the EORTC QLQ-C30 and the EORTC QLQ-CR29 questionnaires, which have been validated in local settings [36,37] were used. Here, higher functional scales (body image, role and emotional functioning) indicated higher quality of life while higher symptoms scale (sexual functioning and financial difficulties) indicated higher symptom burden [50]. Spearman's rank correlations ($r_s$) between selected domain or items in the quality-of-life questionnaire and comparable domains of NeAT-CC questionnaire.

**2.2.7. Floor and ceiling effects.** The sum of all items in each domain was calculated. Domains where >15% of the respondents obtained the lowest or highest possible scores were considered indicative of floor and ceiling effects [51].

## 3. Results

### 3.1. Phase I: Item development

**3.1.1. Item generation and domain identification.** The domains and items that were developed based on the findings from prior qualitative inquiries [20–25] formed the basic components (items) of the needs assessment tool for people living with colorectal cancer. These comprised 67 items, which were grouped into five domains, namely psychosocial, information, practical, financial and employment needs.

**3.1.2. Content validation.** In the first round of content validation, eight items had an item-level content validity index (I-CVI) of less than 0.78, while nine items were considered redundant by the expert panel. After reviewing the comments and the CVI, eight items were dropped from the questionnaire, eight items were rephrased, while another nine items were combined with relevant items to prevent redundancy. The revised questionnaire containing 51 items subsequently underwent a second round of content validation. The overall CVI was 0.96 and the CVIs for each domain ranged from 0.90 to 1.00 as depicted in Table 1. Thus, after two rounds of content validation, the preliminary questionnaire included 51 items that were categorized under six domains namely diagnosis, psychosocial, information, practical, financial and employment.

**Table 1. Content validation of the Needs Assessment Tool for Colorectal Cancer (NeAT-CC).**

| Domain/ Item | Questions | I-CVI | Kappa |
|---|---|---|---|
| **Diagnosis needs (S-CVI=1.00)** | | | |
| A1 | I need healthcare professionals to be understanding and sensitive to my feelings when breaking the news that I have cancer | 1.00 | 1.00 |
| A2 | I need access to counseling and psychological services as soon as possible after the doctor tells me that I have cancer | 1.00 | 1.00 |
| A3 | I need to be informed about cancer support groups as soon as possible | 1.00 | 1.00 |
| **Psychosocial needs (S-CVI=0.90)** | | | |
| B4 | I need healthcare professionals to be more compassionate during my treatment sessions and follow-ups | 1.00 | 1.00 |
| B5 | I need access to counseling and psychological services | 0.83 | 0.82 |
| B6 | I need religious or spiritual support to cope with my emotions | 0.83 | 0.82 |
| B7 | I need help to cope with my worries and fear of recurrence | 0.83 | 0.82 |
| B8 | I need help to accept changes in my body (example: surgery scars, wearing a colostomy bag) and to boost my body image and self confidence | 1.00 | 1.00 |
| B9 | I need my family to give me more emotional support | 1.00 | 1.00 |
| B10 | I need my close family members to receive emotional/psychological support | 1.00 | 1.00 |
| B11 | I need to deal with discrimination in social settings because of my cancer | 0.83 | 0.82 |
| B12 | I need to join a cancer support group | 1.00 | 1.00 |
| **Information needs (S-CVI=0.97)** | | | |
| C13 | I need the information that the doctor gives me to be easily understood | 1.00 | 1.00 |
| C14 | I need the doctor to give me written information about my cancer to read later on | 1.00 | 1.00 |
| C15 | I need adequate information about all the different treatment options before I choose to have them | 1.00 | 1.00 |
| C16 | I need to be informed about my test results as soon as they are known | 0.83 | 0.82 |
| C17 | I need more cancer-related materials (books, brochures, videos, etc.) in the waiting area or online at the hospital website | 1.00 | 1.00 |
| C18 | I need my spouse/family to be given information regarding my cancer | 1.00 | 1.00 |
| C19 | I need information on things that I can do to take better care of myself | 1.00 | 1.00 |
| C20 | I need information about diet to know what I should avoid or what should I be eating more | 1.00 | 1.00 |
| C21 | I need my oncologist to openly discuss traditional and complementary medicine with me | 1.00 | 1.00 |
| C22 | I need the hospital to give traditional and complementary medicine services along with conventional treatment | 0.83 | 0.82 |
| C23 | I need information and help in coping with my sexual difficulties | 1.00 | 1.00 |
| C24 | I need to be informed about fertility issues, and potential solutions, before my treatment | 1.00 | 1.00 |
| **Practical needs (S-CVI=0.93)** | | | |
| D25 | I need to be seen by the same team of doctors at every appointment | 0.83 | 0.82 |
| D26 | I need the waiting time in the hospital to be shortened | 1.00 | 1.00 |
| D27 | I need to be given an explanation if there is a delay in the doctor attending to me | 0.83 | 0.82 |
| D28 | I need help in making appointments and someone to call if I need to change appointments | 1.00 | 1.00 |
| D29 | I need all my hospital appointments to be set on the same day whenever possible | 0.83 | 0.82 |
| D30 | I need to know who to contact if I have any questions or concerns on my disease or treatment in between hospital appointments | 1.00 | 1.00 |
| D31 | I need the hospital facilities and surroundings to be clean, comfortable and pleasant | 1.00 | 1.00 |
| D32 | I need the hospital facilities to be located near each other (example: pharmacy, clinic, payment counter, wards) | 1.00 | 1.00 |
| D33 | I need reserved parking spaces specifically for cancer patients | 1.00 | 1.00 |
| D34 | I need my doctor to manage better the side effects of my cancer and treatments | 1.00 | 1.00 |

*(Continued)*

**Table 1.** (Continued)

| Domain/ Item | Questions | I-CVI | Kappa |
|---|---|---|---|
| D35 | I need my family doctor/ General Practitioner to be also knowledgeable about the side effects of my cancer treatment | 1.00 | 1.00 |
| D36 | I need help to cope with limitations in daily activities/activities that I used to do | 0.83 | 0.82 |
| D37 | I need help to care for my family members who are depending on me (example: children, parents, spouses) | 0.83 | 0.82 |
| **Financial needs (S-CVI:0.98)** | | | |
| E38 | I need information on the costs of my treatments before starting them | 1.00 | 1.00 |
| E39 | I need assistance to pay for my cancer treatments | 1.00 | 1.00 |
| E40 | I need help to understand my insurance benefits and coverage, and in making claims | 1.00 | 1.00 |
| E41 | I need to buy health insurance after my cancer diagnosis | 1.00 | 1.00 |
| E42 | I need guidance and assistance in obtaining financial assistance | 1.00 | 1.00 |
| E43 | I need affordable parking and transportation when I come to the hospital | 1.00 | 1.00 |
| E44 | I need affordable colostomy bags, diapers or wigs | 1.00 | 1.00 |
| E45 | I need help to pay for dietary supplements (example: special milk, special food) | 0.83 | 0.82 |
| E46 | I need affordable special equipment (example: wheelchair, special bed) | 1.00 | 1.00 |
| E47 | I need to pay for hired help at home (example: maid, babysitter) | 1.00 | 1.00 |
| E48 | I need help to cope with a reduced household income due to my illness | 1.00 | 1.00 |
| **Employment needs (S-CVI:1.00)** | | | |
| F49 | I need discrimination at my workplace to be addressed (example: promotion, included in important projects, having job security) | 1.00 | 1.00 |
| F50 | I need workplace flexibility (example: time off for hospital appointments, changes in job scope, flexible hours, work from home) | 1.00 | 1.00 |
| F51 | I need help to find a new job after my cancer diagnosis | 1.00 | 1.00 |

Abbreviations: I-CVI, Item Content Validity Index; S-CVI, Scale Content Validity Index.

**3.1.3. Readability.** The readability of the English items in the NeAT-CC was six, indicating that the items in the questionnaire were easy to be read and understood by students aged around 12 years.

## 3.2. Phase II and III: Scale development and evaluation (Psychometric assessment)

Close to half of the 300 participants who were recruited were Malays (48.3%), followed by the Chinese (40.0%) and Indians (10.3%) (Table 2). Both men and women were equally represented. Three-quarters of the participants were aged above 50 years old. A vast majority received their primary treatment at public hospitals (91.7%). Close to 60% were from low-income households. Approximately, 30% of study participants reported having private health insurance.

A majority of participants have been living with CRC between one and five years (60.3%). Most of them were diagnosed with either stage III (47.7%) or stage IV (34.3%) CRC at initial presentation. Two-thirds of the study participants had undergone colorectal surgery (66.0%), while nearly half had stoma. A high proportion of participants had received cytotoxic chemotherapy.

Of the above, 155 respondents were randomly selected for EFA, while the remaining 145 respondents were included in the CFA.

**3.2.1. Cognitive pre-test and test-retest.** Feedback and comments from cognitive pre-test in 20 patients with CRC indicated that the needs questionnaire was straightforward, easy to understand, comprehensive and relevant. Overall, participants took about 15–20 minutes to complete the questionnaire.

**Table 2. Baseline characteristics of participants in the validation study of NeAT-CC.**

| Characteristic | Test-retest (n = 30) | | EFA and CFA (n = 300) | |
|---|---|---|---|---|
| | n | % | n | % |
| **Gender** | | | | |
| Men | 11 | 36.7 | 152 | 50.7 |
| Women | 19 | 63.3 | 148 | 49.3 |
| **Age (years)** | | | | |
| ≤50 | 12 | 40.0 | 74 | 24.7 |
| 51–64 | 12 | 40.0 | 118 | 39.3 |
| 65–75 | 5 | 16.7 | 100 | 33.3 |
| >75 | 1 | 3.3 | 8 | 2.7 |
| **Marital status** | | | | |
| Single[a] | 5 | 16.7 | 68 | 22.7 |
| Married | 25 | 83.3 | 232 | 77.3 |
| **Ethnicity** | | | | |
| Malay | 13 | 43.3 | 145 | 48.3 |
| Chinese | 14 | 46.7 | 120 | 40.0 |
| Indian | 2 | 6.7 | 31 | 10.3 |
| Others | 1 | 3.3 | 4 | 1.4 |
| **Highest attained education** | | | | |
| Tertiary | 12 | 40.0 | 108 | 36.0 |
| Secondary | 10 | 33.3 | 123 | 41.0 |
| None and Primary | 8 | 26.7 | 69 | 23.0 |
| **Private health insurance ownership** | | | | |
| Yes | 15 | 50 | 85 | 28.3 |
| No | 15 | 50 | 215 | 71.7 |
| **Employment status** | | | | |
| Employed | 5 | 16.7 | 73 | 24.3 |
| Self-employed | 3 | 10.0 | 12 | 4.0 |
| Retired | 9 | 30.0 | 127 | 42.3 |
| Housewife | 7 | 23.3 | 46 | 15.3 |
| Unemployed | 6 | 20.0 | 42 | 14.0 |
| **Monthly household income[b]** | | | | |
| Low (B40) | 16 | 53.4 | 173 | 57.7 |
| Middle (M40) | 10 | 33.3 | 77 | 25.7 |
| High (T20) | 4 | 13.3 | 50 | 16.6 |
| **Primary place of management** | | | | |
| Ministry of Health hospital | 15 | 50.00 | 181 | 60.3 |
| Public university hospital | 8 | 26.7 | 94 | 31.4 |
| Private hospital | 7 | 23.3 | 25 | 8.3 |
| **Comorbidities[c]** | | | | |
| Yes | 9 | 30.0 | 191 | 63.7 |
| No | 21 | 70.0 | 109 | 36.3 |
| **Cancer stage at initial diagnosis** | | | | |
| TNM stage I | 5 | 16.7 | 15 | 5.0 |
| TNM stage II | 5 | 16.7 | 39 | 13.0 |
| TNM stage III | 9 | 30.0 | 143 | 47.7 |
| TNM stage IV | 11 | 36.6 | 103 | 34.3 |

*(Continued)*

**Table 2.** (Continued)

| Characteristic | Test-retest (n = 30) | | EFA and CFA (n = 300) | |
|---|---|---|---|---|
| | n | % | n | % |
| **Time since diagnosis** | | | | |
| Less than 1 year | 9 | 30.0 | 63 | 21.0 |
| 1 to 5 years | 15 | 50.0 | 181 | 60.3 |
| 5 to 10 years | 3 | 10 | 38 | 12.7 |
| Above 10 years | 3 | 10 | 18 | 6.0 |
| **Management of colorectal cancer[d]** | | | | |
| Surgery | 28 | 93.3 | 198 | 66.0 |
| Chemotherapy | 26 | 89.7 | 231 | 77.0 |
| Radiotherapy | 14 | 46.7 | 104 | 34.6 |
| Targeted therapy | 3 | 10.3 | 21 | 7.0 |
| **Presence of stoma** | | | | |
| Yes | 15 | 50.0 | 149 | 49.7 |
| No | 15 | 50.0 | 151 | 50.3 |

Abbreviations: EFA, Exploratory Factor Analysis; CFA, Confirmatory Factor Analysis.

[a]Includes those who were widowed, divorced, or not married.

[b]Based on Malaysian Department of Statistics' Household Income and Basic Amenities Survey 2019.

B40: Bottom 40% (≤ MYR4360), M40: Middle 40% (MYR 4361-MYR 6919),Top 20: Top 20% (≥ MYR6920).

1 USD = 4.40 MYR.

[c]Comorbidities include concurrent medical illnesses such as diabetes mellitus, hypertension, dyslipidaemia and heart diseases.

[d]Not mutually exclusive. Chemotherapy: receiving or completed chemotherapy, radiotherapy: receiving or completed radiotherapy, targeted therapy: receiving or completed targeted therapy.

In the test-retest analysis, all items had a weighted kappa value of above 0.40, indicating moderate to substantial agreement (Table in S1 Table 1). The lowest kappa value was 0.42 for item B6, which states the need for religious or spiritual support to cope with emotions. The highest kappa value was 0.87 for item E40, that mentioned the need to understand insurance benefits and coverage.

**3.2.2. EFA and reliability.** The three items in the employment domain were not included in the EFA as the items were optional questions for participants who were employed or had recently lost their jobs due to their cancer diagnoses. EFA was thus performed with the remaining 48 items. The pattern matrix revealed that items A1 to A3 in the diagnosis needs domain loaded together with the psychosocial needs domain. The diagnosis needs domain was to assess the needs of CRC patients at the point of diagnosis, while psychosocial needs were to assess the needs of CRC survivors over longer periods beyond diagnosis. Thus, items under the diagnosis needs domain were removed from the EFA. The remaining 42 items underwent EFA and results from parallel analysis with polychoric correlations suggested a four-domain model that accounted for 56.6% of the cumulative proportion of variance. Kaiser-Meyer-Olkin (KMO) value of 0.903 and a significant Bartlett's test of sphericity (p-value less than 0.001) were obtained, indicating sampling adequacy. Items E38, C14 and C15 cross-loaded unto two factors, with a difference of less than 0.15 in the loadings and were subsequently dropped from the questionnaire. The domains were named a) psychosocial and information needs (containing 15 items), b) practical and living with cancer needs (8 items), c) healthcare needs (11 items) and d) financial needs (8 items). The final factor loadings from the EFA and the communalities are presented in Table 3. The overall Cronbach's alpha was 0.937, while Cronbach's alpha of the individual domains ranged from 0.763 to 0.891 and CITC ranged from 0.447 to 0.864, indicating excellent internal consistency.

**Table 3. Exploratory factor analysis, Cronbach's alpha and corrected item-total correlation (CITC) of NeAT-CC.**

| Item | Domain/ Items | Loadings | Communalities | CITC |
|------|---------------|----------|---------------|------|
| **1) Diagnosis** (Cronbach's Alpha = 0.763) | | | | |
| A1 | I need healthcare professionals to be understanding and sensitive to my feelings when breaking the news that I have cancer[a] | – | – | 0.583 |
| A2 | I need access to counseling and psychological services as soon as possible after the doctor tells me that I have cancer[a] | – | – | 0.665 |
| A3 | I need to be informed about cancer support groups as soon as possible[a] | – | – | 0.649 |
| **2) Psychosocial and information (**Cronbach's Alpha = 0.898) | | | | |
| B4 | I need healthcare professionals to be more compassionate during my treatment sessions and follow-ups | 0.537 | 0.587 | 0.602 |
| B5 | I need access to counseling and psychological services | 0.615 | 0.521 | 0.622 |
| B6 | I need religious or spiritual support to cope with my emotions | 0.468 | 0.334 | 0.529 |
| B7 | I need help to cope with my worries and fear of recurrence | 0.613 | 0.419 | 0.621 |
| B8 | I need help to accept changes in my body (example: surgery scars, wearing a colostomy bag) and to boost my body image and self confidence | 0.714 | 0.549 | 0.617 |
| B9 | I need my family to give me more emotional support | 0.703 | 0.539 | 0.611 |
| B10 | I need my close family members to receive emotional/psychological support | 0.618 | 0.495 | 0.504 |
| B11 | I need to deal with discrimination in social settings because of my cancer | 0.455 | 0.341 | 0.500 |
| B12 | I need to join a cancer support group | 0.545 | 0.434 | 0.701 |
| C13 | I need the information that the doctor gives me to be easily understood | 0.657 | 0.726 | 0.596 |
| C14 | I need the doctor to give me written information about my cancer to read later on[b] | – | – | – |
| C15 | I need adequate information about all the different treatment options before I choose to have them [b] | – | – | – |
| C16 | I need to be informed about my test results as soon as they are known | 0.550 | 0.620 | 0.606 |
| C17 | I need more cancer-related materials (example: books, brochures, videos, etc.) in the waiting area or online at the hospital website. | 0.400 | 0.449 | 0.622 |
| C18 | I need my spouse/family to be given information regarding my cancer | 0.456 | 0.460 | 0.523 |
| C19 | I need information on things that I can do to take better care of myself | 0.508 | 0.488 | 0.447 |
| C20 | I need information about diet to know what I should avoid or what should I be eating more | 0.547 | 0.542 | 0.631 |
| **3) Healthcare** (Cronbach's Alpha = 0.898) | | | | |
| D25 | I need to be seen by the same team of doctors at every appointment | 0.734 | 0.595 | 0.743 |
| D26 | I need the waiting time in the hospital to be shortened | 0.601 | 0.576 | 0.747 |
| D27 | I need to be given an explanation if there is a delay in the doctor attending to me | 0.623 | 0.489 | 0.457 |
| D28 | I need help in making appointments and someone to call if I need to change appointments | 0.747 | 0.592 | 0.664 |
| D29 | I need all my hospital appointments to be set on the same day whenever possible | 0.675 | 0.502 | 0.712 |
| D30 | I need to know who to contact if I have any questions or concerns on my disease or treatment in between hospital appointments | 0.719 | 0.659 | 0.746 |
| D31 | I need the hospital facilities and surroundings to be clean, comfortable and pleasant | 0.779 | 0.581 | 0.642 |
| D32 | I need the hospital facilities to be located near each other (example: pharmacy, clinic, payment counter, wards) | 0.655 | 0.433 | 0.617 |
| D33 | I need reserved parking spaces specifically for cancer patients who self-drive to the hospital | 0.593 | 0.439 | 0.517 |
| D34 | I need my doctor to manage better the side effects of my cancer and treatments | 0.613 | 0.633 | 0.668 |
| D35 | I need my family doctor/ GP to be also knowledgeable about the side effects of my cancer treatment | 0.688 | 0.592 | 0.633 |
| E43 | I need affordable parking and transportation when I come to the hospital | 0.583 | 0.531 | 0.466 |

*(Continued)*

**Table 3.** (Continued)

| Item | Domain/ Items | Loadings | Communalities | CITC |
|---|---|---|---|---|
| **4) Practical and living with cancer needs** (Cronbach's Alpha = 0.842) | | | | |
| C21 | I need my oncologist to openly discuss traditional and complementary medicine with me | 0.575 | 0.612 | 0.592 |
| C22 | I need the hospital to give traditional and complementary medicine services along with conventional treatment | 0.600 | 0.455 | 0.595 |
| C23 | I need information and help in coping with my sexual difficulties | 0.664 | 0.460 | 0.573 |
| C24 | I need to be informed about fertility issues, and potential solutions, before my treatment | 0.671 | 0.532 | 0.580 |
| D36 | I need help to cope with limitations in daily activities/activities that I used to do | 0.394 | 0.362 | 0.592 |
| D37 | I need help to care for my family members who are depending on me (example: children, parents, spouses) | 0.366 | 0.349 | 0.640 |
| E40 | I need help to understand my insurance benefits and coverage, and in making claims | 0.656 | 0.471 | 0.572 |
| E41 | I need to buy health insurance after my cancer diagnosis | 0.561 | 0.428 | 0.554 |
| **5) Financial** (Cronbach's Alpha = 0.889) | | | | |
| E38 | I need information on the costs of my treatments before starting them[b] | – | | – |
| E39 | I need assistance to pay for my cancer treatments | 0.593 | 0.584 | 0.678 |
| E42 | I need guidance and assistance in obtaining financial assistance | 0.692 | 0.597 | 0.726 |
| E44 | I need affordable colostomy bags, diapers or wigs | 0.558 | 0.395 | 0.505 |
| E45 | I need help to pay for dietary supplements (example: special milk, special food) | 0.884 | 0.757 | 0.864 |
| E46 | I need affordable special equipment (example: wheelchair, special bed) | 0.885 | 0.768 | 0.766 |
| E47 | I need to pay for hired help at home (example: maid, babysitter) | 0.557 | 0.455 | 0.559 |
| E48 | I need help to cope with a reduced household income due to my illness | 0.782 | 0.681 | 0.696 |
| **6) Employment** (Cronbach's Alpha = 0.891) | | | | |
| F49 | I need discrimination at my workplace to be addressed (example: promotion, included in important projects, having job security)[c] | – | – | 0.774 |
| F50 | I need workplace flexibility (example: time off for hospital appointments, changes in job scope, flexible hours, work from home)[c] | – | – | 0.769 |
| F51 | I need help to find a new job after my cancer diagnosis[c] | – | – | 0.837 |

Abbreviations: CITC, Corrected Item-Total Correlation.

[a]Item A1 to A3 were removed from the EFA as the items were loading together with items in the psychosocial domain that applied only to patients who were newly diagnosed.

[b]Items C14, C15 and E38, were removed due to cross-loading more than 0.15.

[c]Items F49 to F51 represent employment domains that were only answered by those working groups. Items were removed from EFA but were retained due to expert justification on the importance of the items.

**3.2.3. CFA.** The 42 items from EFA, three items from diagnosis needs, and three items from employment needs were included for CFA. Results from the CFA using PLS-SEM are shown in Fig 1. The factor loadings of all items exceeded 0.50 (Table in S2 Table 2), with three domains having an AVE of more than 0.50, while another three domains had an AVE of 0.40 (Table 4). The newly developed tool also demonstrated good internal consistency with a CR value of above 0.80 for all the reflective domains, and a Cronbach's alpha that was larger than 0.70. The HTMT ratios were less than 0.90 for all the outer model domains (Table in S3 Table 3), with no cross-loadings between the items, demonstrating good discriminant validity.

**3.2.4. Criterion validity.** The correlation between scales in the quality of life questionnaire (EORTC QLQ-C30 and QLQ CR-29) and the related items in NeAT-CC are presented in Table 5. Significant negative correlations were observed

**Fig 1. Confirmatory factor analysis model of the needs assessment tool for colorectal cancer (NeAT-CC) using partial least square-structural equation modelling.**

**Table 4. Measurement model assessment of NeAT-CC from confirmatory factor analysis.**

| Domain | Number of items | Cronbach's Alpha | Composite Reliability (CR) | Average Variance Extracted (AVE) |
|---|---|---|---|---|
| Diagnosis need | 3 | 0.774 | 0.871 | 0.693 |
| Psychosocial and information need | 15 | 0.907 | 0.921 | 0.439 |
| Healthcare need | 12 | 0.885 | 0.905 | 0.443 |
| Practical and living with cancer need | 8 | 0.794 | 0.847 | 0.411 |
| Financial need | 7 | 0.881 | 0.907 | 0.553 |
| Employment | 3 | 0.774 | 0.868 | 0.687 |

Abbreviations: CR, Composite Reliability; AVE, Average Variance Extracted.

**Table 5. Correlation (rs) between NeAT-CC domain and EORTC QLC C30 and EORTC QLQ CR29 domains.**

| NeAT-CC domain (items) | Criterion Validity (Quality of life) | $r_s$ | p-value |
|---|---|---|---|
| **Total need score** | Global Health Status[b] (QLQ C-30) | −0.36 | <0.001 |
| | Social functioning[b] (QLQ C-30) | −0.33 | <0.001 |
| | Financial difficulties[a] (QLQ C-30) | 0.42 | <0.001 |
| | Anxiety[b] (QLQ CR-29) | −0.30 | <0.001 |
| **Psychosocial and information need** | Global Health Status[b] (QLQ C-30) | −0.27 | <0.001 |
| | Financial difficulties[a] (QLQ C-30) | 0.31 | <0.001 |
| | Social functioning[b] (QLQ C-30) | −0.28 | <0.001 |
| **Practical and living with cancer** | Global Health Status[b] (QLQ C-30) | −0.35 | <0.001 |
| | Social functioning[b] (QLQ C-30) | −0.32 | <0.001 |
| | Sexual symptoms: Male[a] (QLQ CR-29) | 0.30 | <0.001 |
| **Financial need** | Global Health Status[b] | −0.42 | <0.001 |
| | Role Functioning[b] (QLQ C-30) | −0.32 | <0.001 |
| | Financial difficulties[a] (QLQ-C30) | 0.60 | <0.001 |
| | Anxiety[b] (QLQ CR-29) | −0.30 | <0.001 |

Abbreviations: Questionnaire–Core 30; EORTC QLQ-CR29, European Organisation for Research and Treatment of Cancer Quality of Life Questionnaire–Colorectal Module.

Footnote:

[a]Symptom scales, a high score on the symptoms scale indicates a high level of symptomatology/ problems.

[b]Functional scale, high score indicate better quality of life.

between the total needs score, financial needs, and the practical and living with cancer needs domain of the NeAT-CC with the global health status of the quality of life questionnaire, with rs = −0.36 (p < 0.001), rs = −0.42 (p < 0.001), and rs = −0.30 (p < 0.001) respectively.

A significant positive correlation between the financial domain of the need's questionnaire and the financial difficulties scale from the quality of life questionnaire was also observed with $r_s$ = 0.60 and p value <0.001. This indicated that high financial needs positively correlated with higher financial difficulties. Practical and living with cancer needs (with items related to sex and fertility from the NeAT-CC) also positively correlated with sexual symptoms scores for males with $r_s$ = 0.30 (p < 0.001). Overall, the NeAT-CC demonstrated adequate criterion validity as depicted in Table 5.

**3.2.5. Floor and ceiling effects.** The mean score of the domains ranged from 2.77 to 3.94. With the exception of the diagnosis needs domain with a ceiling effect of 21.3%, no floor and ceiling effects were observed for the other domains (Table 6).

**Table 6. Response distribution for each domain of the NeAT-CC (floor and ceiling effect).**

| Domain | Mean Score (SD) | Lowest Score (floor, %) | Highest Score (ceiling, %) |
|---|---|---|---|
| Diagnosis need | 3.51 (1.37) | 3.0 | 21.3 |
| Psychosocial and information need | 3.75 (0.96) | 0.3 | 5.3 |
| Healthcare need | 3.94 (0.94) | 0.3 | 11 |
| Practical and living with cancer | 2.67 (1.21) | 2.0 | 3.0 |
| Financial need | 2.82 (1.46) | 4.3 | 5.3 |
| Employment[a] | 2.77 (1.46) | 8.6 | 10.5 |

Abbreviations: SD, Standard Deviation.

[a]Employment needs were assessed only among those who were still working or recently lost their job due to cancer (n=105).

### 3.3. Comparison of NeAT-CC with other commonly used needs assessment tools

We undertook a post-hoc comparison of NeAT-CC against other commonly used needs assessment tools (Table 7). Besides being a tool, that specifically targeted individuals living with and beyond CRC, the NeAT-CC was noted to stand out in its comprehensiveness where the needs around the time of diagnosis could also be measured in newly diagnosed individuals. Compared to previous tools, it included items measuring financial as well as employment needs.

## 4. Discussion

Our questionnaire development and validation exercise resulted in a 48-item bilingual Needs Assessment Tool for Colorectal Cancer (NeAT-CC) encompassing six domains, namely, diagnosis needs, psychosocial and information needs, healthcare needs, practical and living with cancer needs, financial needs, and employment needs. Study findings suggest that the NeAT-CC is a psychometrically sound tool with low respondent burden that can be used to assess the needs of people living with CRC and beyond.

Apart from being guided by findings of prior qualitative inquiries in local settings, the co-design approach that we had adopted ensured that the perspectives of a wide range of stakeholders including patients and caregivers were taken into account. Compared to prior questionnaires, more items related to financial and employment needs were included in the newly developed NeAT-CC, in line with expert recommendations [52,53]. The NeAT-CC questionnaire possesses distinct features that set it apart from commonly used questionnaires, including catering to patients who are newly diagnosed to assess their needs around the time of diagnosis, and covering a comprehensive array of needs following CRC along the cancer care continuum including financial needs, employment needs, needs related to traditional and complementary medicine, fertility needs, religious and spiritual support, needs related to diet, and needs for medical devices such as stoma bags.

The high proportion of younger individuals with colorectal cancer (< 50 years) [54], high risk of financial toxicity following cancer diagnosis [31] and high employment needs amid limited employment protection policies [21] in our settings, necessitated the inclusion of these domains. Moreover, the needs relating to traditional and complementary medicine that were covered in our tool reflect the cultural beliefs of our multi-ethnic population. It is worth noting that to the best of our knowledge, none of the currently available needs assessment tools were particularly developed for people living with and beyond CRC. Furthermore, only the Chinese version of SCNS-SF questionnaire was cross culturally validated among patients with CRC in Hong Kong and Taiwan [55]. All of the above strengthen our optimism that the newly developed NeAT-CC will be both meaningful and relevant to the needs of people living with and beyond CRC, not only in the local setting but also in other multi-ethnic settings.

The items in the NeAT-CC were finalized using multiple techniques, namely factor analysis, test-retest and confirmatory factor analysis. The test–retest estimate of reliability was determined by assessing the correlation between

**Table 7. Comparison of commonly used needs assessment tools in individuals with (colorectal) cancer.**

| Questionnaire | Needs Assessment Tools for people living with Colorectal Cancer (NeAT-CC) | Supportive Care Needs Survey-Short Form 34 (SCNS-SF34) | Comprehensive needs assessment tool in cancer (CNAT) | Cancer Survivors' Unmet Needs Measure (CaSUN) |
|---|---|---|---|---|
| Year of launch | 2022 | 2009 | 2011 | 2007 |
| Origin | Malaysia | Australia | Korea | Australia |
| Target population | Colorectal cancer | All cancers | All cancers | All cancers |
| Settings | Outpatient and inpatient | Not available | Outpatient and inpatient | Outpatient |
| Number of items | 48 | 34 | 51 | 28 |
| Domains | Six domains:<br>i) diagnosis<br>ii) psychosocial and information (one item on religious or spiritual needs)<br>iii) healthcare needs<br>iv) practical and daily living (items on sexual, fertility and complementary treatment)<br>v) financial<br>vi) employment | Five domains;<br>i) psychological<br>ii) health system and information<br>iii) physical and daily living<br>iv) patient care and support<br>v) sexuality | Seven domains:<br>i) healthcare staff<br>ii) physical symptoms (items on sexual life)<br>iii) psychological problems<br>iv) information (one items on alternative treatment and financial)<br>v) social/religious/spiritual support<br>vi) practical support<br>vii) hospital facilities and services (one item on employment and return to work) | Five domains:<br>i) existential survivorship (one item on spiritual belief)<br>ii) comprehensive care<br>iii) information<br>iv) quality of life and<br>v) relationships (include one item on sex life) |
| Response options | 0: no need<br>1: very low need<br>2: low need<br>3: moderate need<br>4: high need<br>5: very high need | 1: no need; not applicable<br>2: no need; satisfied)<br>3: low need<br>4: moderate need<br>5: high need | Likert 0–10 scale<br>(0 = not at all and 10 = a great deal) | Indicate for each item<br>(a) no unmet need/not applicable or<br>(b) if they do experience a need, how strong the need is either(weak/moderate/strong) |
| Scoring | Subscales scores | The total score and subscales scores | Total sum of item scores | Total needs, total unmet needs |
| Administration | Self-administered | Self-administered | Self-administered | Self-administered |
| Time to complete | 15min | 10min | Not available | 10 min |
| Unique attributes | 1. Designed for people living with and beyond colorectal cancer<br>2. Assesses needs at diagnosis for newly diagnosed patients.<br>3. Includes items on<br>  i. financial needs<br>  ii. employment needs<br>  iii. needs related to traditional and complementary medicine<br>  iv. fertility needs<br>  v. religious and spiritual support<br>  vi. needs related to diet<br>  vii. medical devices/equipment (stoma bag and wheelchair) | 1. Cover needs that are common to all cancer patients throughout their cancer journey.<br>2. Cross-culturally adapted in many settings<br>3. Multiple language versions including English, Spanish, French, Japanese, Traditional Chinese, Mandarin, Dutch and German. | 1. Developed and validated in South Korea (Asian)<br>2. Cover needs that are common to all cancer patients throughout their cancer journey.<br>3. Includes items on<br>  i. needs related to traditional and complementary/alternative treatment<br>  ii. religious support | 1. Focused on patients living beyond cancer (1–15 years after diagnosis)<br>2. A domain on quality of life with two items on management of side effects, and changes to quality of life<br>3. Include item on needs related to diet |

sets of scores at different time points [56] and was found to be particularly reliable for self-reported surveys [57]. Moderate to substantial reliability was observed in our study, surpassing previous tools with lower reliability [7] or no test–retest data [58].

Following EFA, three items that were removed were related to the need for written information (C14), information about all treatment options (C15), and information on the costs of treatment (E38). All three items cross-loaded with healthcare needs, suggesting they were likely highly correlated and complex [43]. The two domains that were excluded from EFA (diagnosis and employment) were retained in CFA due to their importance [20,21,23] and expert justification [43]. The diagnosis needs assess needs at diagnosis, highlighting the persistent emotional toll of colorectal cancer during treatment and long-term survivorship [22]. The employment needs domain, meanwhile, reflects the needs among the working group, which are crucial for understanding the challenges faced by the working class, including economic impacts and the ability to return to work post-treatment. It is also noteworthy that CFA demonstrated that these two domains achieved adequate convergent and discriminant validity to justify their retention in the NeAT-CC.

Quality of life assessment is commonly used to identify patients' concerns and wellbeing. However, it only assesses the presence and severity of a concern in an individual and not whether they need assistance to address their concerns [14]. Comparatively, needs assessment identifies the range of concerns experienced by patients for which they require intervention or additional help [27]. Despite the differences, multiple domains of needs have been shown to be predictive of health-related quality of life measures, a finding that is also corroborated by prior research [44]. With no gold standard for needs assessment, criterion validity, which measures the relationship between a test score and a relevant measure, was established by correlating NeAT-CC with the EORTC QLQ-C30 and QLQ-CR29 quality of life questionnaires. While further validation with a gold standard is recommended [19], the significant correlation with quality of life demonstrates NeAT-CC's effectiveness in capturing the relationship between needs and overall well-being.

The ceiling effects for the diagnosis needs domain indicated that most survivors were experiencing very high levels of needs. Although presence of ceiling effects may impair the responsiveness of the scale and may indicate limited content validity [51], our findings are in line with prior evidence where similar (floor and ceiling) effects were observed [45]. It was postulated that these observations may not necessarily be due to limitation of the scale but rather because people living with and beyond cancer were having either very high needs or very low care needs in some domains [59], such as diagnosis needs [20].

In sum, our newly developed NeAT-CC is not only psychometrically sound but also straightforward and easily understandable for patients. Given that the tool may be self-administered, it is expected that its use in routine clinical practice will pose minimal administrative burden on health systems. Utilization of the NeAT-CC in clinical and research settings will undoubtedly yield valuable knowledge and evidence on the needs of people living with and beyond CRC that can be leveraged to guide the development of new services that are patient-centered, or optimization of existing supportive and survivorship care services. Importantly, the tool may also be used by cancer civil societies to guide and evaluate their programs.

To the best of our knowledge, this is the first needs assessment tool that has been developed exclusively for CRC, which spans the supportive and survivorship care needs along the cancer care continuum. We undertook a rigorous methodology to develop and validate the NeAT-CC, in accordance with international standards [18]. Here, it is noteworthy that almost two-thirds of prior studies in this area did not employ CFA as part of their analyses [10]. Furthermore, the inclusion of patients from multiple settings allowed for diversity in ethnic and socioeconomic backgrounds, where the baseline characteristics of the study participants, including gender, ethnicity, cancer stage, and insurance status, are broadly consistent with national colorectal cancer (CRC) data and recent studies. A higher CRC burden among males and advanced-stage diagnoses was also observed in this study [60]. The predominance of Malay participants aligns with previously reported trends [54]. Additionally, 28.3% of participants reported having personal health insurance, which is comparable to national estimates [61].

The diversity informed the inclusion of additional items relevant to the health practices of multicultural societies, such as the need for traditional and complementary therapies. It is therefore believed that the NeAT-CC has potential for wider application and can be cross-culturally validated for use in other settings with diverse populations. However, in

high-income countries with comprehensive social support systems, certain domains such as employment and access to treatment may require modification to ensure contextual relevance.

In the local context, future studies should validate the NeAT-CC among the indigenous populations, as well as Mandarin-and Tamil-speaking patients, to ensure that their voices are also represented. Future plans also include implementing a cohort study to assess supportive care needs longitudinally and to compare these needs between urban and rural populations.

## 5. Conclusion

The NeAT-CC is expected to enable accurate and effective needs assessment following CRC in Malaysia, with potential for adaptation in other multi-ethnic and/ or low- to middle-income settings. This, in turn, may facilitate the prioritization and development of supportive and survivorship care services that meet the needs of people living with and beyond CRC, as well as guide the allocation of scarce health resources.

## Supporting information

**S1 Table. Test-retest reliability results of the Needs Assessment Tool for Colorectal Cancer (NeAT-CC).** This table presents the test–retest reliability analysis for the Needs Assessment Tool for Colorectal Cancer (NeAT-CC), including Weighted Kappa for each item.
(DOCX)

**S2 Table. 2 Confirmatory factor analysis of the needs assessment tool for colorectal cancer (NeAT-CC).** This table presents factor loadings for each item across the six domains.
(DOCX)

**S3 Table. Heterotrait–monotrait (HTMT) ratios of the domains for discriminant validity.** This table displays the HTMT ratios between domains of the NeAT-CC to demonstrate discriminant validity.
(DOCX)

## Acknowledgments

We would like to acknowledge Ms Kelly Ming-Ying Lai and Dr Murallitharan Munisamy for their contribution as part of the expert panel during the co-design phase of the study. We also thank Mr Muhd Ash-Shafawi Adznan from Hospital Selayang and Ms Wong Chung Heong from the Colorectal Cancer Survivorship Society Malaysia (CORUM) for their assistance in identifying potential study participants. The authors would like to thank the Director-General of the Ministry of Health Malaysia for his permission to publish this article.

## Author contributions

**Conceptualization:** Nur Nadiatul Asyikin Bujang, Nirmala Bhoo-Pathy.

**Data curation:** Nur Nadiatul Asyikin Bujang, Yek Ching Kong, Awang Bulgiba, Mahmoud Danaee, Nirmala Bhoo-Pathy.

**Formal analysis:** Nur Nadiatul Asyikin Bujang.

**Methodology:** Nur Nadiatul Asyikin Bujang, Awang Bulgiba, Mahmoud Danaee, Nirmala Bhoo-Pathy.

**Project administration:** Nur Nadiatul Asyikin Bujang, Yek Ching Kong, Muthukkumaran Thiagarajan, April Camilla Roslani, Muhammad Radzi Abu Hassan, Matin Mellor Abdullah, Mehesinder Singh, Wan Zamaniah Wan Ishak, Awang Bulgiba, Nirmala Bhoo-Pathy.

**Supervision:** Awang Bulgiba, Mahmoud Danaee, Nirmala Bhoo-Pathy.

**Validation:** Nur Nadiatul Asyikin Bujang.

**Writing – original draft:** Nur Nadiatul Asyikin Bujang, Nirmala Bhoo-Pathy.

**Writing – review & editing:** Nur Nadiatul Asyikin Bujang, Yek Ching Kong, Muthukkumaran Thiagarajan, April Camilla Roslani, Muhammad Radzi Abu Hassan, Matin Mellor Abdullah, Mehesinder Singh, Wan Zamaniah Wan Ishak, Awang Bulgiba, Mahmoud Danaee, Nirmala Bhoo-Pathy.

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
