## [Decision Letter · Decision Letter 0]

4 Dec 2024

PONE-D-24-35504DEVELOPMENT AND VALIDATION OF A DUAL LANGUAGE NEEDS ASSESSMENT TOOL FOR PEOPLE LIVING WITH COLORECTAL CANCER (NeAT-CC)PLOS ONE

Dear Dr. Bhoo-Pathy,

Thank you for submitting your manuscript to PLOS ONE. After careful consideration, we feel that it has merit but does not fully meet PLOS ONE’s publication criteria as it currently stands. Therefore, we invite you to submit a revised version of the manuscript that addresses the points raised during the review process. Reviewers have highlighted significant issues that need to be addressed.

We look forward to receiving your revised manuscript.

Kind regards,

Ali Haider Mohammed

Academic Editor

PLOS ONE

2. In the online submission form, you indicated that [The datasets used and analyzed during the current study are not publicly available due to some information may compromised participants privacy but are available from the corresponding author on reasonable request.]. All PLOS journals now require all data underlying the findings described in their manuscript to be freely available to other researchers, either 1. In a public repository, 2. Within the manuscript itself, or 3. Uploaded as supplementary information. This policy applies to all data except where public deposition would breach compliance with the protocol approved by your research ethics board. If your data cannot be made publicly available for ethical or legal reasons (e.g., public availability would compromise patient privacy), please explain your reasons on resubmission and your exemption request will be escalated for approval.

Additional Editor Comments (if provided):

Reviewers' comments:

Reviewer's Responses to Questions

**Comments to the Author**

1. Is the manuscript technically sound, and do the data support the conclusions?

Reviewer #1: Yes

Reviewer #2: Partly

2. Has the statistical analysis been performed appropriately and rigorously? 

Reviewer #1: Yes

Reviewer #2: Yes

3. Have the authors made all data underlying the findings in their manuscript fully available?

Reviewer #1: No

Reviewer #2: Yes

4. Is the manuscript presented in an intelligible fashion and written in standard English?

Reviewer #1: Yes

Reviewer #2: Yes

5. Review Comments to the Author

Reviewer #1: Thank you for the opportunity to provide peer review for the article “Development and validation of a dual language needs assessment tool for people living with colorectal cancer (NeAT-CC)”.

The well written article focuses on creating a dual language needs assessment tool for people living with or after colorectal cancer in a multi-ethnic and resource limited country. The authors argue that previous needs assessment tools were mostly developed for affluent countries and that so far there are no tools specifically validated for colorectal cancer in low- to mid-income countries. To create their needs assessment tool, the authors followed rigorous methodological guidelines. This includes the COnsensus-based Standards for the selection of health status Measurement INstruments (COSMIN) checklist, validations with factor analyses and a later correlation of the NeAT-CC with known quality of life questionnaires.

Major points:

• Statistics: Did you test for other factor models for the confirmatory factor analysis (CFA)?

• Are there any specific plans in the future with the newly validated NeAT-CC tool, how you would bring it into practice, who could be analysing the questionnaires and implement the changes? Are there any cut-offs of the patient scores that help with its interpretation?

Minor points:

• Please state more in detail, why other already available assessment tools are not applicable in Malaysia. Where are the differences (better or worse)? What exactly is missing in the other tools and which problems would arise if the other tools are applied in your population? Could the NeAT-CC tool instead be used also for high-income countries? Or if not, what are the possible problems when applying the NeAT-CC in those countries.

• In the results of the abstract it is not clear which are the six domains of needs. “(…) namely diagnosis, psychosocial and information, healthcare, practical and living with cancer, financial and employment.”

• Is the study population representative of the general population? In 2.2.1 it is described, that some people were excluded that were unfit or couldn’t converse (maybe add a flow-chart to clarify). There should be more information if the baseline characteristics of the studied patients match with the general population of Malaysia (percentage of private insurance, income, etc. as in Table 2).

• Abbreviations should be stated in every Table.

• It would be interesting to have a longitudinal study to see, whether the data are reproducible over time and if the NeAT-CC is able to show changes in needs of colorectal cancer patients over time. This not only for the individual itself but also generally for the whole population, if new services and programs are implemented based on the needs evaluations with the NeAT-CC.

• I could not read Figure 1.

The study uses robust statistics to address an important area. It can, with some refinements, potentially make a significant impact in the field of survivorship care.

Reviewer #2: Dear Authors,

I appreciate the effort and dedication invested in your research. Your work addresses an important area of study, and I believe it has significant potential to contribute to the field. However, some amendments are necessary to enhance the clarity and overall quality of the manuscript. Below are my recommendations:

Introduction

The introduction could benefit from additional focus and elaboration on the motivation for the study. Specifically, the statement:

"Prior reviews also suggest that many of the existing tools do not comprehensively cover all domains of needs following cancer, such as social needs, or do not demonstrate adequate psychometric properties. While certain needs following CRC have been reported to be similar to other cancers, there are some unique challenges, such as dealing with stoma that necessitates the development of a disease-specific tool."

This section is critical in establishing the importance and relevance of your work. I recommend expanding on these points to provide greater detail and context. Doing so will help clarify the motivation for your study and underscore its significance.

Methods

The inclusion criteria require further elaboration. Please provide additional details to ensure that the reader fully understands the selection process and the rationale behind it. A clear description of inclusion criteria enhances the transparency and reproducibility of your research.

Discussion

The discussion should focus more on explaining your current results rather than making extensive comparisons with other assessment tools or delving into their details. While such comparisons are valuable, they might be better suited for the literature review or introduction. Keeping the discussion centered on interpreting your findings will strengthen the narrative and highlight the contributions of your study.

Closing Remarks

Thank you for your dedication to this important topic. I hope these suggestions will assist you in refining your manuscript. I look forward to seeing the revised version.

Best regards,

6. PLOS authors have the option to publish the peer review history of their article (what does this mean? ). If published, this will include your full peer review and any attached files.

**Do you want your identity to be public for this peer review?** For information about this choice, including consent withdrawal, please see our Privacy Policy .

Reviewer #1: **Yes: ** Roman Adam

Reviewer #2: **Yes: ** Bassam Abdul Rasool Hassan

---

## [Author Response · Author response to Decision Letter 1]

8 Aug 2025

Respond to reviewer [PONE-D-24-35504]

Reviewer #1:

Thank you for the opportunity to provide peer review for the article “Development and validation of a dual language needs assessment tool for people living with colorectal cancer (NeAT-CC)”.

The well written article focuses on creating a dual language needs assessment tool for people living with or after colorectal cancer in a multi-ethnic and resource limited country. The authors argue that previous needs assessment tools were mostly developed for affluent countries and that so far there are no tools specifically validated for colorectal cancer in low- to mid-income countries. To create their needs assessment tool, the authors followed rigorous methodological guidelines. This includes the COnsensus-based Standards for the selection of health status Measurement INstruments (COSMIN) checklist, validations with factor analyses and a later correlation of the NeAT-CC with known quality of life questionnaires.

Response: We sincerely appreciate your constructive feedback. We will make every effort to address all the comments thoroughly in our revised manuscript.

Major points:

• Statistics: Did you test for other factor models for the confirmatory factor analysis (CFA)?

Response: Thank you for the comment. We did not test alternative factor structures during the CFA, as the analysis was guided by the factor structure identified through the preceding exploratory factor analysis (EFA). According to our methodological approach, the CFA was conducted to confirm the dimensional structure derived from the EFA results. To address this point, a few sentences have been added to subsection 2.2.6 (Confirmatory Factor Analysis) on page 12 of the revised manuscript.

Confirmatory factor analysis was conducted via the SmartPLS software 3.2.9 ..." During the CFA phase, we evaluated the factor structure established through EFA without contrasting alternative models. This decision was predicated on the robust theoretical framework and empirical precision of the EFA solution. The EFA demonstrated a clear and interpretable structure characterized by high item loadings and minimal cross-loadings, which informed the development of the CFA model."

• Are there any specific plans in the future with the newly validated NeAT-CC tool, how you would bring it into practice, who could be analysing the questionnaires and implement the changes? Are there any cut-offs of the patient scores that help with its interpretation?

Response: Yes, several plans are underway to facilitate the practical application of the newly validated NeAT-CC tool. Future plans include translating the questionnaire into additional local languages, such as Tamil and Mandarin, and conducting the survey among Indigenous Yes populations to ensure inclusivity across Malaysia’s multiethnic population.

We appreciate the reviewer’s suggestion. This point has already been addressed in the manuscript. Kindly refer to Page 36, Paragraph 3, where the following sentence is included:

“In the local context, future studies should validate the NeAT-CC among the indigenous populations, as well as Mandarin-and Tamil-speaking patients, to ensure that their voices are also represented”

Furthermore, we also plan to conduct the study in rural settings and implement a cohort study to assess supportive care needs longitudinally and compare needs across urban and rural populations. A corresponding sentence has been added on page 37:

“Future plans also include implementing a cohort study to assess supportive care needs longitudinally and to compare these needs between urban and rural populations.”

As for the cut-off scores, there is currently no standardized threshold to differentiate levels of need. At this stage, we propose the use of latent standardized scores generated from the PLS-SEM model for research purposes. For clinical applicability, we are in the process of developing categorical thresholds (e.g., low, moderate, high need) to simplify interpretation and facilitate use by healthcare professionals. The questionnaire can be administered by trained health staff or researchers, and the findings can be used to inform patient-centered care planning, resource allocation, and referrals for psychosocial or supportive services.

Minor points:

• Please state more in detail, why other already available assessment tools are not applicable in Malaysia. Where are the differences (better or worse)? What exactly is missing in the other tools and which problems would arise if the other tools are applied in your population? Could the NeAT-CC tool instead be used also for high-income countries? Or if not, what are the possible problems when applying the NeAT-CC in those countries.

Response: We thank the reviewer for the comment. This point has been addressed in the discussion section (Page 33 and Page 36), where the differences between the NeAT-CC and existing tools are explained.

Page 33, Paragraph 2; “The NeAT-CC questionnaire possesses distinct features that set it apart from commonly used questionnaires, including catering to patients who are newly diagnosed to assess their needs around the time of diagnosis, and covering a comprehensive array of needs following CRC along the cancer care continuum including financial needs, employment needs, needs related to traditional and complementary medicine, fertility needs, religious and spiritual support, needs related to diet, and needs for medical devices such as stoma bags.”

Page 36, Paragraph 2; “To the best of our knowledge, this is the first needs assessment tool that has been developed exclusively for CRC, which spans the supportive and survivorship care needs along the cancer care continuum. We undertook a rigorous methodology to develop and validate the NeAT-CC, in accordance with international standards [15]. Here, it is noteworthy that almost two-thirds of prior studies in this area did not employ CFA as part of their analyses [6].”

To address the reviewer’s comment on “Could the NeAT-CC tool also be used in high-income countries, we would like to highlight that this point has been addressed earlier in the manuscript, Page 36 Paragraph 2;

“It is therefore believed that the NeAT-CC has potential for wider application and can be cross-culturally validated for use in other settings with diverse populations.”

To further clarify the applicability of the tool in high-income countries, we also added:

Page 37, Paragraph 2; “However, in high-income countries with comprehensive social support systems, certain domains such as employment and access to treatment may require modification to ensure contextual relevance.”

• In the results of the abstract it is not clear which are the six domains of needs. “(…) namely diagnosis, psychosocial and information, healthcare, practical and living with cancer, financial and employment.”

Response: Thank you for the comment. To improve clarity, Roman numerals have been added in the abstract to clearly differentiate the six domains of needs. The revised sentence now reads: “...encompassed six domains of needs, namely (i) diagnosis, (ii) psychosocial and information, (iii) healthcare, (iv) practical and living with cancer, (v) financial, and (vi) employment.”

• Is the study population representative of the general population? In 2.2.1 it is described, that some people were excluded that were unfit or couldn’t converse (maybe add a flow-chart to clarify). There should be more information if the baseline characteristics of the studied patients match with the general population of Malaysia (percentage of private insurance, income, etc. as in Table 2).

Response: We sincerely thank the reviewer for this thoughtful comment regarding the representativeness of the study population and the suggestion to include a flowchart. While we appreciate the recommendation, we respectfully believe that a flowchart may not be essential in this context, as the recruitment process was relatively straightforward.

The inclusion and exclusion criteria are described in Section 2.2.1, where the inclusion criteria sentence was revised for clarity (see Page 9); " Malaysians above the age of 18 years who were living with and beyond CRC, diagnosed with any stage of CRC at least one month prior to recruitment, with or without stoma, were recruited.”

To further clarify representativeness, we have added the following sentence, Paragraph 1, Page 9: “ Malay is the national language and is widely spoken across all major ethnic groups in Malaysia, while English serves as the second language and is commonly used in healthcare and research settings.”

Additionally, to ensure diversity, respondents were recruited from multiple sites across Klang Valley including a public university hospital, Ministry of Health hospitals, private hospital, and a non-governmental organization to capture a broad range of ethnic and socioeconomic backgrounds. This is detailed in Paragraph 2, Page 9 of the revised manuscript.

Thank you too for your valuable comment regarding the need for more information on the representativeness of the study population. To address this, we have added the following sentence in the Discussion section, Page 36, Paragraph 1;

“... where the baseline characteristics of the study participants, including gender, ethnicity, cancer stage, and insurance status, are broadly consistent with national colorectal cancer (CRC) data and recent studies. A higher CRC burden among males and advanced-stage diagnoses was also observed in this study [60]. The predominance of Malay participants aligns with previously reported trends [54]. Additionally, 28.3% of participants reported having personal health insurance, which is comparable to national estimates [61] ”

• Abbreviations should be stated in every Table.

Response: Thank you for the comment. Abbreviations have been added to the legend of each relevant table in accordance with journal guidelines. Please refer to the following:

Table 1: Abbreviations: I-CVI, Item Content Validity Index; S-CVI, Scale Content Validity Index (please refer page 17).

Table 2: Abbreviations: EFA, Exploratory Factor Analysis; CFA, Confirmatory Factor Analysis (please refer page 21).

Table 3: Abbreviations: CITC, Corrected Item-Total Correlation (please refer page 26).

Table 4: Abbreviations: CR, Composite Reliability; AVE, Average Variance Extracted (please refer page 28).

Table 5: Abbreviations: Questionnaire–Core 30; EORTC QLQ-CR29, European Organisation for Research and Treatment of Cancer Quality of Life Questionnaire–Colorectal Module (please refer page 29).

Table 6: Abbreviations: SD, Standard Deviation (please refer page 30).

• It would be interesting to have a longitudinal study to see, whether the data are reproducible over time and if the NeAT-CC is able to show changes in needs of colorectal cancer patients over time. This not only for the individual itself but also generally for the whole population, if new services and programs are implemented based on the needs evaluations with the NeAT-CC.

Response: Thank you for the insightful suggestion. We recognise the value of assessing supportive care needs over time and are planning a longitudinal cohort study as part of our future research efforts. This has been reflected in the revised manuscript. A sentence has been included on page 37: “Future plans also include implementing a cohort study to assess supportive care needs longitudinally and to compare these needs between urban and rural populations.”

• I could not read Figure 1.

Figure 1 is included in the main text; please refer to page 27

The study uses robust statistics to address an important area. It can, with some refinements, potentially make a significant impact in the field of survivorship care.

Response: Thank you. We appreciate the reviewer’s comment

Reviewer #2:

Dear Authors,

I appreciate the effort and dedication invested in your research. Your work addresses an important area of study, and I believe it has significant potential to contribute to the field. However, some amendments are necessary to enhance the clarity and overall quality of the manuscript. Below are my recommendations:

Response: We appreciate the reviewer’s thoughtful comment.

Introduction

The introduction could benefit from additional focus and elaboration on the motivation for the study. Specifically, the statement:

"Prior reviews also suggest that many of the existing tools do not comprehensively cover all domains of needs following cancer, such as social needs, or do not demonstrate adequate psychometric properties. While certain needs following CRC have been reported to be similar to other cancers, there are some unique challenges, such as dealing with stoma that necessitates the development of a disease-specific tool."

This section is critical in establishing the importance and relevance of your work. I recommend expanding on these points to provide greater detail and context. Doing so will help clarify the motivation for your study and underscore its significance.

Response: Thank you for this insightful comment. We have expanded on the points raised to provide greater detail and context regarding the importance and relevance of our work. Specifically, we have revised the Introduction section (Page 5, Paragraph 3) to include the following:

”Commonly used tools such as the Supportive Care Need Survey (SCNS-SF) [6], Cancer Survivors’ Unmet Needs Measure (CaSUN) [7], Survivors’ Unmet Needs Survey (SUNS) [8], and the Comprehensive Needs Assessment Tool in Cancer (CNAT) developed in Korea [9] were reviewed.“

And Page 6,Paragraph 1

“Needs assessment tools tailored to these unique challenges are essential to ensure that survivors’ concerns are effectively identified and addressed [14].”

Methods

The inclusion criteria require further elaboration. Please provide additional details to ensure that the reader fully understands the selection process and the rationale behind it. A clear description of inclusion criteria enhances the transparency and reproducibility of your research.

Response: The inclusion criteria were revised (Page 9): " Malaysians above the age of 18 years who were living with and beyond CRC, diagnosed with any stage of CRC at least one month prior to recruitment, with or without stoma, were recruited.”

Discussion

The discussion should focus more on explaining your current results rather than making extensive comparisons with other assessment tools or delving into their details. While such comparisons are valuable, they might be better suited for the literature review or introduction. Keeping the discussion centered on interpreting your findings will strengthen the narrative and highlight the contributions of your study.

Response: We thank the reviewer for the comment. In response, we revised the manuscript by removing the comparative sentence regarding the NeAT-CC and other available tools from Page 33, Paragraph 2. This content has now been incorporated into the Introduction section (Page 5, Paragraph 3) to better align with the manuscript’s structure.

Closing Remarks

Thank you for your dedication to this important topic. I hope these suggestions will assist you in refining your manuscript. I look forward to seeing the revised version.

Response: We thank the reviewer for the valuable feedback provided on our manuscript.

---

## [Decision Letter · Decision Letter 1]

8 Sep 2025

DEVELOPMENT AND VALIDATION OF A DUAL LANGUAGE NEEDS ASSESSMENT TOOL FOR PEOPLE LIVING WITH COLORECTAL CANCER (NeAT-CC)

PONE-D-24-35504R1

Dear Dr.Nirmala,

We’re pleased to inform you that your manuscript has been judged scientifically suitable for publication and will be formally accepted for publication once it meets all outstanding technical requirements.

Kind regards,

Ali Haider Mohammed

Academic Editor

PLOS ONE

Additional Editor Comments (optional):

Reviewer #2:

Reviewers' comments:

Reviewer's Responses to Questions

**Comments to the Author**

1. If the authors have adequately addressed your comments raised in a previous round of review and you feel that this manuscript is now acceptable for publication, you may indicate that here to bypass the “Comments to the Author” section, enter your conflict of interest statement in the “Confidential to Editor” section, and submit your "Accept" recommendation.

Reviewer #2: All comments have been addressed

2. Is the manuscript technically sound, and do the data support the conclusions?

Reviewer #2: Yes

3. Has the statistical analysis been performed appropriately and rigorously? 

Reviewer #2: Yes

4. Have the authors made all data underlying the findings in their manuscript fully available?

Reviewer #2: Yes

5. Is the manuscript presented in an intelligible fashion and written in standard English?

Reviewer #2: Yes

6. Review Comments to the Author

Reviewer #2: Dear Authors,

I really appreciate your hard work.

All the required amendments were addressed in a very professional way.

Regards,

7. PLOS authors have the option to publish the peer review history of their article (what does this mean? ). If published, this will include your full peer review and any attached files.

**Do you want your identity to be public for this peer review?** For information about this choice, including consent withdrawal, please see our Privacy Policy .

Reviewer #2: **Yes: ** Bassam Abdul Rasool Hassan

---

## [Editor Report · Acceptance letter]

PONE-D-24-35504R1

PLOS ONE

Dear Dr. Bhoo-Pathy,

I'm pleased to inform you that your manuscript has been deemed suitable for publication in PLOS ONE. Congratulations! Your manuscript is now being handed over to our production team.

Kind regards,

on behalf of

Dr. Ali Haider Mohammed

Academic Editor

PLOS ONE